# Seeded X-ray free-electron laser generating radiation with laser statistical properties

Oleg Yu. Gorobtsov[1,9], Giuseppe Mercurio[2,10], Flavio Capotondi[3], Petr Skopintsev[1,11], Sergey Lazarev [1,4], Ivan A. Zaluzhnyy [1,5,12], Miltcho B. Danailov[3], Martina Dell'Angela[6], Michele Manfredda[3], Emanuele Pedersoli [3], Luca Giannessi[3,7], Maya Kiskinova [3], Kevin C. Prince [3,8], Wilfried Wurth[1,2] & Ivan A. Vartanyants [1,5]

The invention of optical lasers led to a revolution in the field of optics and to the creation of such fields of research as quantum optics. The reason was their unique statistical and coherence properties. The emerging, short-wavelength free-electron lasers (FELs) are sources of very bright coherent extreme-ultraviolet and X-ray radiation with pulse durations on the order of femtoseconds, and are presently considered to be laser sources at these energies. FELs are highly spatially coherent to the first-order but in spite of their name, behave statistically as chaotic sources. Here, we demonstrate experimentally, by combining Hanbury Brown and Twiss interferometry with spectral measurements that the seeded XUV FERMI FEL-2 source does indeed behave statistically as a laser. The results may be useful for quantum optics experiments and for the design and operation of next generation FEL sources.

[1] Deutsches Elektronen-Synchrotron DESY, Notkestrasse 85, D-22607 Hamburg, Germany. [2] Department of Physics, University of Hamburg and Center for Free Electron Laser Science, Luruper Chausse 149, D-22761 Hamburg, Germany. [3] Elettra-Sincrotrone Trieste, 34149 Basovizza (Trieste), Italy. [4] National Research Tomsk Polytechnic University (TPU), pr. Lenina 30, 634050 Tomsk, Russia. [5] National Research Nuclear University MEPhI (Moscow Engineering Physics Institute), Kashirskoe shosse 31, 115409 Moscow, Russia. [6] CNR- IOM Istituto Officina dei Materiali, 34149 Trieste, Italy. [7] ENEA C.R. Frascati, Via E. Fermi 45, 00044 Frascati, Rome, Italy. [8] Molecular Model Discovery Laboratory, Department of Chemistry and Biotechnology, School of Science Swinburne University of Technology, Melbourne, VIC 3122, Australia. [9] Present address: Department of Materials Science and Engineering, Cornell University, Ithaca, NY 14850, USA. [10] Present address: European XFEL GmbH, Holzkoppel 4, D-22869 Schenefeld, Germany. [11] Present address: Laboratory for Biomolecular Research, Division of Biology and Chemistry, Paul Scherrer Institute, PSI Aarebrucke, 5232 Villigen, Switzerland. [12] Present address: Department of Physics, University of California San Diego, La Jolla CA 92093, USA. Correspondence and requests for materials should be addressed to I.A.V. (email: ivan.vartaniants@desy.de)

Glauber in his pioneering work[1] stated that a truly coherent source of radiation should be coherent in all orders of intensity correlation functions. Most sources of radiation in the optical wavelength range behave statistically as thermal or chaotic sources. Optical lasers, due to their significantly different radiation properties, provide unique opportunities in science and technology. As first demonstrated in the time domain, single-mode or phase-locked optical lasers are not only coherent in the first-order, but also in the second-order of intensity correlation functions[2,3]. This is also valid in the spatial domain and distinguishes laser sources from chaotic sources of radiation. This difference is especially important for quantum optics experiments that are extended now to classical fields[4,5], in which high-order coherence properties of the source play an important role. The possibility of employing similar properties of sources in the extreme-ultraviolet (XUV) and X-ray range with extremely high brightness is the main attraction for completely new and exciting applications of free-electron lasers (FELs).

Most of the presently operating short-wavelength FELs[6–10] generate radiation using the self-amplified spontaneous emission (SASE) process[11], where the radiation is produced stochastically by the electron bunch shot noise. As a result, the spatial and especially temporal structure of the X-ray pulse fluctuates strongly from shot to shot. Typically, each SASE pulse contains a large number of longitudinal modes without any phase locking between them. As such, SASE FELs are spatially highly coherent sources in first-order (with the degree of spatial coherence reaching 80%[12,13]), but from a statistical point of view, they behave as chaotic sources of radiation[14–16]. The first steps have been taken towards exploiting the first-order coherence of FELs[17,18], but to date, no consistent high-order statistical measurements have been performed at any seeded FEL.

FERMI is the first seeded single-pass FEL in the XUV regime and hosts two sources[7] FEL-1 and FEL-2. Recent measurements have demonstrated that FERMI FEL-1 is coherent in the temporal domain[19,20]. The question naturally arises whether this indicates only first-order coherence, or whether the FERMI FEL is also coherent in higher orders, thus satisfying the definition of Glauber. This is an important question, as it defines the potential of this light source for experiments based on high-order intensity correlations, a common requirement for quantum optics investigations[3].

Here we employ Hanbury Brown and Twiss (HBT) interferometry to explore the statistical properties of radiation from a FEL source. We perform the higher-order intensity correlation measurements at FERMI and demonstrate that it statistically behaves as a laser source.

## Results

**HBT correlation and spectral analysis.** The basic idea behind HBT interferometry[21,22] is to extract statistical properties of radiation from the normalised second-order intensity correlation function[23]

$$g^{(2)}(\boldsymbol{r}_1, \boldsymbol{r}_2) = \langle I(\boldsymbol{r}_1) I(\boldsymbol{r}_2) \rangle / \langle I(\boldsymbol{r}_1) \rangle \langle I(\boldsymbol{r}_2) \rangle, \qquad (1)$$

where $I(\boldsymbol{r}_1)$ and $I(\boldsymbol{r}_2)$ are intensities at different spatial positions measured simultaneously, and the brackets $\langle \dots \rangle$ denote averaging over a large ensemble of different radiation pulses. The statistical behaviour of the $g^{(2)}$ correlation function is fundamental in quantum optics[3] and strongly depends on the radiation type. For example, for coherent laser sources $g^{(2)}$ is equal to one[2,3] but for chaotic sources it behaves quite differently (see Methods Eq. (4)).

Measurements were performed at the DiProI end-station of FERMI using the FEL-2 source and acquiring simultaneously data from the on-line spectrometer PRESTO (see Fig. 1, Experimental details, and Supplementary Note 1). This feature of FERMI allowed us to analyse simultaneously the spectral and intensity profiles of each pulse delivered and measured at the DiProI end-station. One more important feature of FERMI is that its operation can be switched from seeded to SASE mode[24].

We first analysed the X-ray intensity distribution of the FEL pulses in the seeded and SASE regimes of FERMI FEL-2 by employing HBT correlation analysis (Eq. (1)). Intensity correlation functions measured in the vertical and horizontal directions for both operation modes are shown in Fig. 2a–d. The contrast values of the intensity correlation function (see Methods Eq. (3)) in the central part of the beam were on the order of 0.03–0.04 for the seeded beam and slightly higher about 0.1–0.15 for the SASE. The remarkable difference between the two regimes of operation becomes evident by correlating these observations to the corresponding spectral profiles.

The average spectrum, both in the seeded and SASE regimes of FERMI operation, was approximately Gaussian in shape but with a quite different relative bandwidth $\Delta\omega/\omega_0$ (see Fig. 2e, f). In the seeded regime it was about $4.6 \times 10^{-4}$ and in SASE $6 \times 10^{-3}$, an order of magnitude broader. The low values of the contrast measured in the seeded regime, in the case of a chaotic Gaussian beam, would indicate the presence of at least 25–30 independent longitudinal modes[14–16]. However, from our on-line spectrometer measurements, we observed from one to five modes varying from pulse to pulse, with one mode usually dominating. At the same time, the values of contrast obtained in SASE mode match the number of observed spectral modes (about 10), supporting the assumption of chaotic character of the source in the SASE regime (see Methods Eq. (4)).

Next, we proceeded with analysis of the spectral dependence of the $g^{(2)}$ function in the seeded regime of FERMI operation by implementing a sorting procedure (see Supplementary Note 2). Two data sets of pulses were then considered: $10^3$ pulses with the largest contribution of the main mode and $10^3$ pulses with the smallest contribution. Intensity correlation

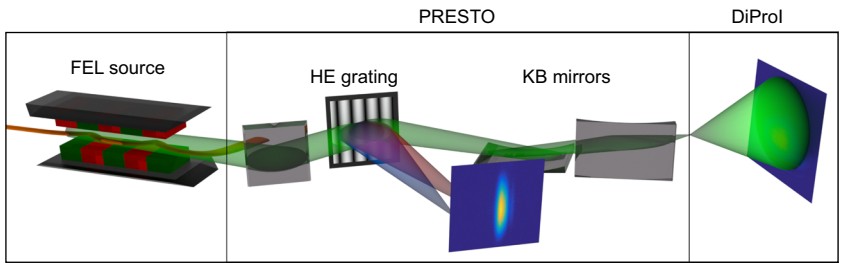

**Fig. 1** Schematic layout of the experiment. Radiation generated in the undulators is focused by Kirkpatrick–Baez mirrors and the detector is installed out of focus to observe the direct beam. Radiation from each pulse is partially diffracted by a grating to the spectrometer detector to observe on-line pulse spectra

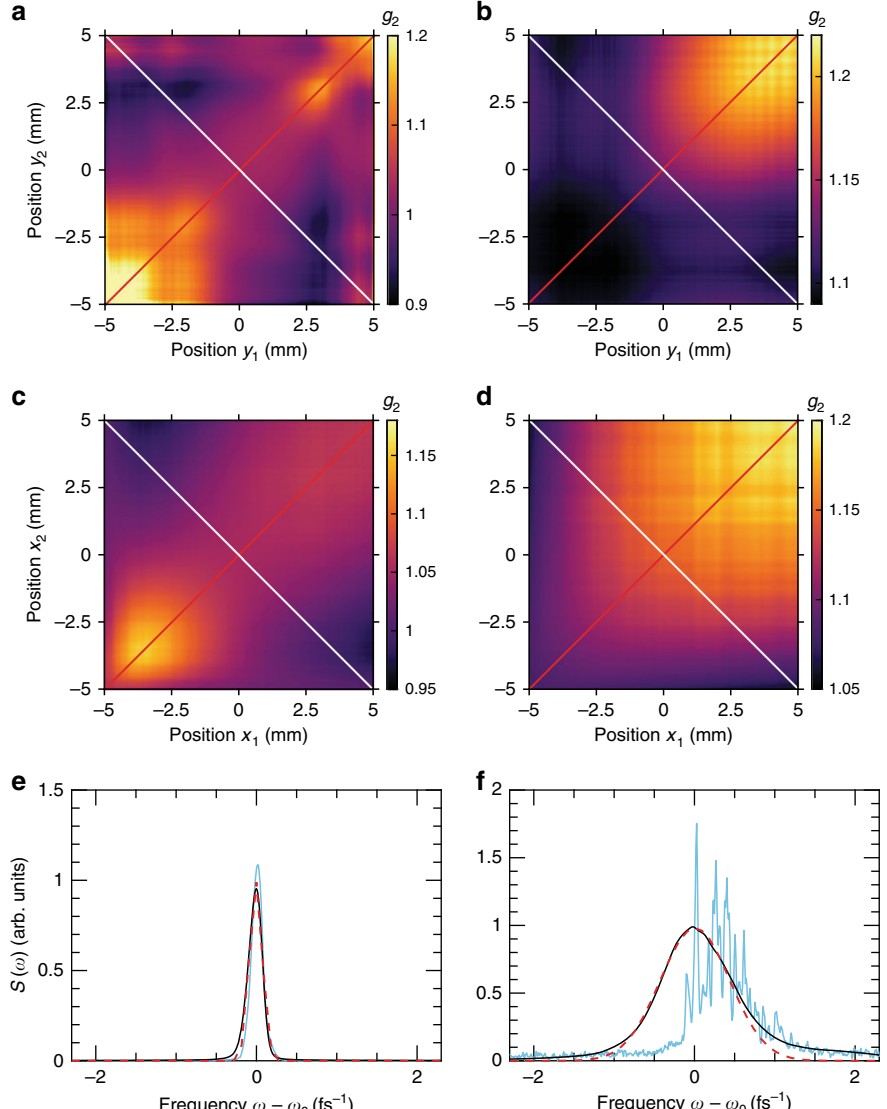

**Fig. 2** Correlation functions and spectra measured in different modes of operation. **a–d** Intensity correlation functions for seeded (**a**, **c**) and self-amplified spontaneous emission (SASE) (**b**, **d**) regimes of operation in vertical (**a**, **b**) and horizontal (**c**, **d**) directions. Spectral structure of an individual pulse (blue line), average spectrum (black line) and Gaussian fit (red dashed line) for seeded (**e**) and SASE (**f**) regimes of operation

functions determined by Eq. (1) for these two data sets are presented in Fig. 3a, c and examples of the single and multimode pulses are shown in Fig. 3b, d. Remarkably, these two data sets produce similar correlation functions with low values of contrast of about 0.02 and 0.07 for the data set with the smallest and largest contribution of the main mode, respectively. Based on these results, we conclude that the seeded FERMI FEL-2 source is not behaving as a chaotic source but rather as a laser source, even in the case when several modes are present in the spectrum. This implies that the spectral modes are potentially phase locked, as in the case of FERMI FEL-1[19,20], which is an important finding for coherent control experiments[17].

**Probability distribution and dispersion values of intensity**. To further investigate the difference between the seeded and SASE operation modes, the probability distribution of total intensity was studied in both cases (see Fig. 4a, c). As known from the first-order coherence theory, there is little difference between a multimode laser without phase locking and a chaotic source[25]. In both cases the total pulse intensity distribution follows a Gamma distribution. We indeed observed such behaviour in the SASE regime of FERMI with about eight longitudinal modes (see Fig. 4c). At the same time, in the seeded mode, fitting data with a Gamma distribution does not well represent the measured distribution and gives the unlikely result of 58 modes (see Fig. 4a), contradicting the spectral observation. A much better fit was provided by a Gaussian density function, with a number of modes close to one[25] (see Methods Eqs. (5) and (6)). This is also consistent with the statistical behaviour of the FERMI FEL-2 source conforming to that of a phase-locked laser.

We also analysed the dispersion values of the total intensity $I$ given by $\zeta_{tot} = \langle \delta I^2 \rangle / \langle I \rangle^2$, where $\delta I = I - \langle I \rangle$. They were measured at the spectrometer as a function of the radiation bandwidth for both regimes of operation (see Fig. 4b, d and Supplementary Note 3). In the SASE mode (Fig. 4d), we observed the typical behaviour measured also at other SASE FEL sources[14,15] with saturation at the narrower bandwidth values. We found that this saturation value was smaller (0.5) than the expected value of unity. Such behaviour was

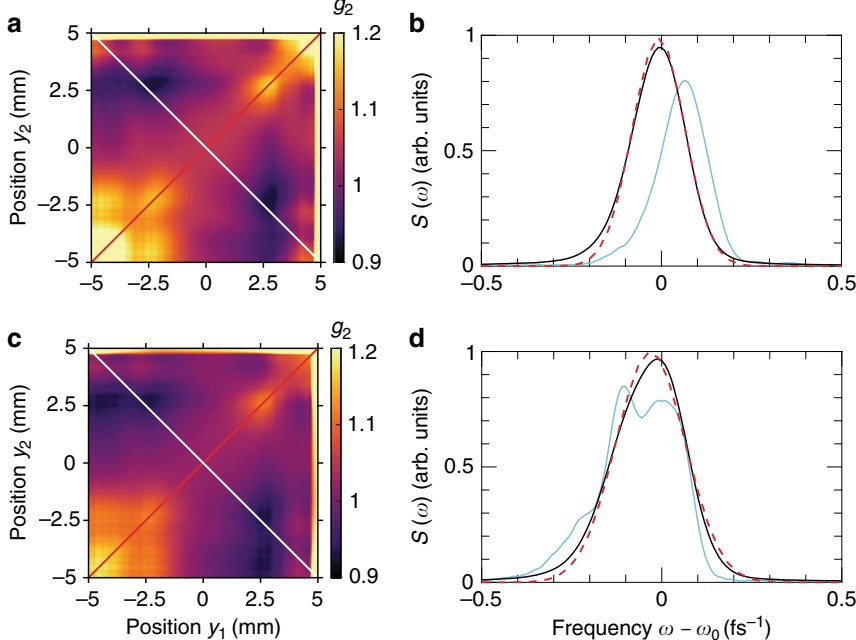

**Fig. 3** Seeded mode single and multiple pulses analysis. Intensity correlation functions (**a**, **c**) and spectral structure (**b**, **d**) for the sorted pulses with the lowest (**a**, **b**) and highest (**c**, **d**) number of modes. In (**b**, **d**) representative single pulses are shown by blue lines, an average over $10^3$ pulses is shown by black lines, and Gaussian fit by red dashed lines. Note the different scale in frequency $\omega$ in comparison with Fig. 2b, d

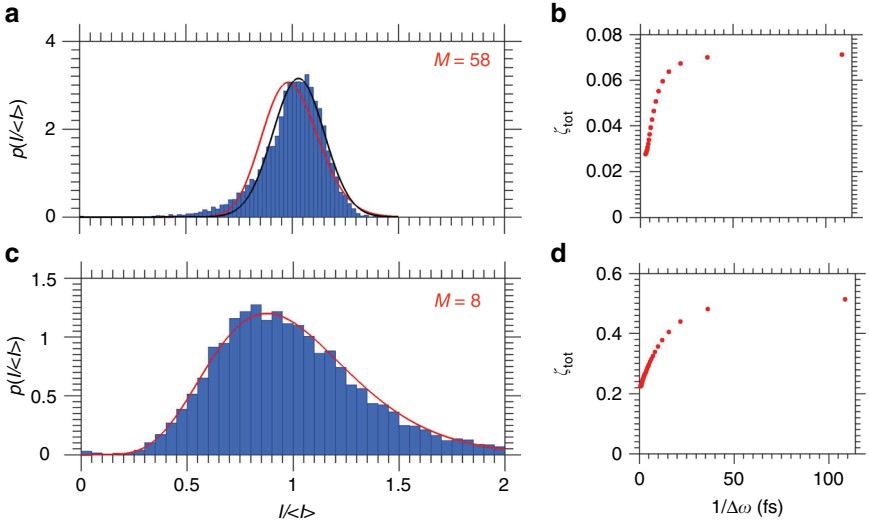

**Fig. 4** Histograms and dispersion values of intensity in different modes of operation. **a**, **c** Histograms of probability distributions of intensity, p(I/⟨I⟩), for seeded (**a**) and self-amplified spontaneous emission (SASE) (**c**) regimes. The red line represents a fit by a Gamma function and the black line represents a fit by a Gaussian function. Number of modes determined from the Gamma distribution is indicated by M. (**b**, **d**) Contrast behaviour as a function of inverse bandwidth for seeded (**b**) and SASE (**d**) regimes of operation

observed previously[15] and may be explained by the limited spectrometer resolving power (see Supplementary Note 4). In the seeded mode, we observed similar behaviour of the dispersion values (see Fig. 4b), but the saturation value corresponding to a narrow bandwidth was much lower, about 0.07.

## Discussion

For coherent laser radiation we expect the $g^{(2)}$ function to be equal to one in the whole spatial and spectral range[3]. The residual contribution of 2–7% at FERMI may be attributed to a combination of incoherent sources of noise and variations of external parameters, such as electron beam energy or trajectory. Even

if stabilised by feedbacks, these parameters are subject to shot-to-shot variations affecting the properties of the emitted light. Such fluctuations do not affect the coherence of individual pulses, but may affect the average value in Eq. (1). Among genuine chaotic contributions, a residual contribution from micro-bunching instability may be mixing noise with the coherent energy modulation of the seed (see Supplementary Note 4). In a seeded FEL, the statistical fluctuations of the seed laser itself are translated to the XUV by the harmonic conversion process. Values of $g^{(2)}$ higher than one are often observed in optical lasers[26], caused by mixing of chaotic radiation or quantum noise with the coherent laser radiation.

A combination of HBT interferometry and spectral measurements allowed us to demonstrate that a seeded FEL is fundamentally different in its statistical properties from a SASE-based FEL. These measurements are a decisive step forward in understanding the basic properties of FELs. While SASE FELs behave statistically as chaotic sources, the seeded FERMI FEL is equivalent in its statistical properties to a coherent laser in the definition of Glauber. Importantly, the degeneracy parameter (number of photons in a single mode) for the seeded FEL FERMI reaches a number as high as $10^{11}$–$10^{12}$. This is by two orders of magnitude higher than in the case of SASE FELs[14], where a monochromator has to be used to pass a single longitudinal mode[18].

By performing high-order correlation analysis we foresee that a range of quantum optics experiments previously explored with optical fields may be performed with X-rays. For example, the Hong–Ou–Mandel effect[27], in which the quantum interference of indistinguishable photons is more intense than that of classical waves, should be observable with the light from FERMI. Coincidence detection by two detectors (second-order intensity correlation measurements) may resolve a still open question on diffraction of stimulated emission with intense FEL light[28]. The application of ideas and methods based on high-order coherence at X-ray energies is in its early stage of development[4,5], and the knowledge that second-order coherent FEL light is available at FERMI permits the design and execution of next generation experiments, that strongly rely on high-order statistical properties of the radiation. An open and intriguing question regards the statistical properties of self-seeded FELs[29,30]. In contrast to an externally seeded FEL, this perfectly first-order coherent source may show different second-order statistical properties.

## Methods

**Correlation functions.** The normalised first-order correlation function is defined as[3,25]

$$g^{(1)}(\mathbf{r_1}, \mathbf{r_2}) = \langle E^*(\mathbf{r_1})E(\mathbf{r_2})\rangle / \left\langle \sqrt{I(\mathbf{r_1})} \right\rangle \left\langle \sqrt{I(\mathbf{r_2})} \right\rangle, \quad (2)$$

where $E(\mathbf{r_1})$ and $E(\mathbf{r_2})$ are complex amplitudes of the wave field at different spatial positions measured simultaneously, and the brackets $\langle \dots \rangle$ denote averaging over a large ensemble of different radiation pulses. The first-order correlation function (2) represents the mutual intensity function[25].

The normalised second-order intensity correlation function $g^{(2)}(\mathbf{r_1}, \mathbf{r_2})$ is defined by Eq. (1). An important quantity derived from the second-order intensity correlation function (1) is the contrast defined as

$$\zeta_2(\mathbf{r}) = g^{(2)}(\mathbf{r}, \mathbf{r}) - 1. \quad (3)$$

Chaotic sources with Gaussian statistics may be described by the following intensity correlation function[14,31]

$$g^{(2)}(\mathbf{r_1}, \mathbf{r_2}) = 1 + \zeta_2(D_\omega)\left|g^{(1)}(\mathbf{r_1}, \mathbf{r_2})\right|^2. \quad (4)$$

Here $\zeta_2(D_\omega)$ is the contrast function defined in (3), which in the case of chaotic sources depends strongly on the radiation frequency bandwidth $D_\omega$ and $g^{(1)}(\mathbf{r_1}, \mathbf{r_2})$ is the first-order correlation function (2). In the case of a chaotic pulsed beam, $\zeta_2(D_\omega)$ is determined by $\tau_c/T$, where $\tau_c = 2\pi/D_\omega$ is the coherence time and $T$ is the pulse duration of the FEL radiation[31]. In this limit the number of longitudinal modes $M_t$ defined as $M_t = T/\tau_c$ is inversely proportional to the contrast function $\zeta_2(D_\omega)$. Notice also that for a perfect chaotic Gaussian source $g^{(2)}(\mathbf{r}, \mathbf{r}) = 1 + \zeta_2(D_\omega)$ and does not depend on position $\mathbf{r}$.

**Experimental details.** We performed the experiment at the beamline DiProI of the FERMI using the FEL-2 source with both seeded and SASE modes. The scheme of the experiment is shown in Fig. 1. The double cascade source FEL-2 tuned to the wavelength of 10.9 nm and average power of 11 μJ per pulse was used to generate seeded and SASE radiation. The radiation was focused with a Kirkpatrick–Baez optical system, and the intensity distribution was detected at a distance of 0.5 m from the focus (transmission of the beamline was about 65–70% at 10.9 nm wavelength). The beam divergence, after the refocusing optics, was about 1.0 mrad. An in-vacuum Andor Ikon CCD detector with 2048 × 2048 pixels of 13.5 × 13.5

μm² size was used for intensity measurements of the direct beam. A small portion of the beam was partially diffracted by a grating to the shot-by-shot PRESTO spectrometer[32]. This diagnostic provides a spectrum of each pulse simultaneously with the measurements at the DiProI end station. Several series of both direct beam and spectral images at different FEL parameters were recorded. Each series consisted of $10^4$ shots, measured with 10 Hz frequency.

The spectrometer is characterised by a resolving power of $1.8 \times 10^4$ at 10.9 nm[32], corresponding to the separation of two spectral lines of about $6 \times 10^{-4}$ nm. This implies that the spectrometer resolving power is sufficient to correctly measure the width of the seeded FEL pulses ($5 \times 10^{-3}$ nm). The SASE spectrum contains spikes resulting from the spectral superposition of uncorrelated temporal spikes separated in time. The duration of the temporal spikes can be derived from the SASE average spectral width ($6.5 \times 10^{-2}$ nm) and corresponds to about 6 fs. The finest structure in the spectral distribution depends on maximum temporal separation between the spikes. Assuming this separation to be of the order of 1 ps, i.e. the duration of the electron current (see Supplementary Note 4) we calculate the width of these finest structures in the spectrum to be about $4 \times 10^{-4}$ nm, which would require a resolving power of about $3 \times 10^{-4}$ to be observable. For this reason we found that the saturation value in Fig. 4d was smaller (0.5) than the expected value of unity. At the maximum spectral resolution of the spectrometer, it may still collect about two longitudinal modes which gives the contrast value in Eq. (3) of about 0.5.

**Intensity distribution for chaotic and laser sources.** In the case of a chaotic source obeying Gaussian statistics, the probability $p(I)$ that the total intensity of the pulse takes value $I$, follows a Gamma distribution[25,33]

$$p(I) = \frac{M^M}{\Gamma(M)}\left(\frac{I}{\langle I\rangle}\right)^{M-1} \exp\left(-M\frac{I}{\langle I\rangle}\right), \quad (5)$$

where $M$ is the number of degrees of freedom, or modes, and $\langle I\rangle$ is an average intensity of the pulse.

In the case of laser radiation, the probability distribution of the total intensity may be described as originating from various noise effects and obeys a normal distribution[25]

$$p(I) = \frac{1}{\sqrt{2\pi\sigma^2}}\exp\left(-\frac{(I - \langle I\rangle)^2}{2\sigma^2}\right), \quad (6)$$

where $\sigma$ is the width of the distribution.

## Data availability

All relevant data are available from the corresponding author.

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

## Acknowledgements

We thank the whole staff of FERMI for their support and excellent operation of the facility as well as for the fruitful discussions during this experiment; especially we are thankful to G. Penco and G. De Ninno from the FERMI facility. We acknowledge E. Weckert for helpful discussions and support of the project as well as fruitful discussions with F. Kärtner and careful reading of the manuscript by Yu. N. Obukhov and T. Laarmann.

## Author contributions

I.A.V., W.W., K.C.P. and M.K. conceived the experiment and coordinated the experimental efforts. L.G. planned FERMI FEL-2 operation. M.K., F.C., M.M. and E.P. prepared the DiProI end-station for measurements. O.Yu.G., G.M., F.C., P.S., S.L., I.Z., M.D., M.D.'A, M.M., E.P., M.K., K.C.P., W.W. and I.A.V. performed the measurements. O.Yu.G. and P.S. analysed the data. O.Yu.G, I.A.V., L.G. and K.C.P. wrote the manuscript with contributions and improvements from all authors. All authors read and discussed the manuscript.

## Additional information

**Competing interests:** The authors declare no competing interests.

