## [Peer Review File · Nature Communications]

REVIEWERS' COMMENTS:

Reviewer #5 (Remarks to the Author):

I have read the paper after the new revisions and the reply to the referees' concerns. In my opinion, the very strong statement that the manuscript opens the way for experiments on quantum optics at x-ray wavelengths is misleading. While the main result of paper, which is the demonstration that the seeded Fermi XFEL shows a second order intensity correlation function of a coherent laser is interesting and important, there is still a long way for experiments on quantum optics, and for many experiments on quantum optics the correlation function is not important, so the results of the present work is irrelevant for them. I do not think that the main challenge that prevents the performance of experiments at x-ray wavelengths is the lack of knowledge of the second order correlation function. The discussion that the authors added in response to the comments of the referees is not useful. It is too general, misleading, and explanation about the contribution of the current work to quantum optics is vague.

Here is reply to referee #5 comments

Reviewer #5 (Remarks to the Author):

I have read the paper after the new revisions and the reply to the referees' concerns. In my opinion, the very strong statement that the manuscript opens the way for experiments on quantum optics at x-ray wavelengths is misleading. While the main result of paper, which is the demonstration that the seeded Fermi XFEL shows a second order intensity correlation function of a coherent laser is interesting and important, there is still a long way for experiments on quantum optics, and for many experiments on quantum optics the correlation function is not important, so the results of the present work is irrelevant for them. I do not think that the main challenge that prevents the performance of experiments at x-ray wavelengths is the lack of knowledge of the second order correlation function. The discussion that the authors added in response to the comments of the referees is not useful. It is too general, misleading, and explanation about the contribution of the current work to quantum optics is vague.

Our reply:

To follow reviewer suggestion we removed reference to future quantum optics experiments at FERMI from the Abstract part of the manuscript. It was already removed from the conclusion part on earlier stage of the review process. By that we satisfied all recommendations of the reviewer #5.